# Comparing State-of-the-Art Deep Learning Algorithms for the Automated Detection and Tracking of Black Cattle

**DOI:** 10.3390/s23010532

**Published:** 2023-01-03

**Authors:** Su Myat Noe, Thi Thi Zin, Pyke Tin, Ikuo Kobayashi

**Affiliations:** 1Interdisciplinary Graduate School of Agriculture and Engineering, University of Miyazaki, Miyazaki 889-2192, Japan; 2Graduate School of Engineering, University of Miyazaki, Miyazaki 889-2192, Japan; 3Field Science Center, Faculty of Agriculture, University of Miyazaki, Miyazaki 889-2192, Japan

**Keywords:** cattle detection, cattle tracking, deep learning, multi-object tracking, precision livestock farming, re-identification

## Abstract

Effective livestock management is critical for cattle farms in today’s competitive era of smart modern farming. To ensure farm management solutions are efficient, affordable, and scalable, the manual identification and detection of cattle are not feasible in today’s farming systems. Fortunately, automatic tracking and identification systems have greatly improved in recent years. Moreover, correctly identifying individual cows is an integral part of predicting behavior during estrus. By doing so, we can monitor a cow’s behavior, and pinpoint the right time for artificial insemination. However, most previous techniques have relied on direct observation, increasing the human workload. To overcome this problem, this paper proposes the use of state-of-the-art deep learning-based Multi-Object Tracking (MOT) algorithms for a complete system that can automatically and continuously detect and track cattle using an RGB camera. This study compares state-of-the-art MOTs, such as Deep-SORT, Strong-SORT, and customized light-weight tracking algorithms. To improve the tracking accuracy of these deep learning methods, this paper presents an enhanced re-identification approach for a black cattle dataset in Strong-SORT. For evaluating MOT by detection, the system used the YOLO v5 and v7, as a comparison with the instance segmentation model Detectron-2, to detect and classify the cattle. The high cattle-tracking accuracy with a Multi-Object Tracking Accuracy (MOTA) was 96.88%. Using these methods, the findings demonstrate a highly accurate and robust cattle tracking system, which can be applied to innovative monitoring systems for agricultural applications. The effectiveness and efficiency of the proposed system were demonstrated by analyzing a sample of video footage. The proposed method was developed to balance the trade-off between costs and management, thereby improving the productivity and profitability of dairy farms; however, this method can be adapted to other domestic species.

## 1. Introduction

Cattle are one of the most commonly raised livestock and a primary protein source for people across numerous cultures and geographic regions. Managing the health of cattle improves productivity and is ideally performed by monitoring individual cows. Moreover, behavior has been well-established as a strong indicator of cattle health. However, manually monitoring the behavior of individual cows is not practical or sustainable with the existing staffing and workflow.

Ranches and dairy farms often contain many similar cattle, which are difficult to distinguish visually. With the development of modern information and automation technology, automated cattle monitoring has become practical. With improvements in sensor technology and wireless network technology, researchers have adapted traditional solutions into numerous methods including: mechanical, electronic, and biometric solutions. Unfortunately, these methods are disadvantageous in many ways. For example, ear tags cause stress on the cattle and may also be lost or damaged over time. Radiofrequency identification (RFID) technology is also commonly used to track cattle. This technology requires active RFID tags attached to the cattle, tag readers, and radio communication between them, thus requiring a high initial cost and negatively impacting the cattle’s health and well-being. The purpose of our future research is to detect and analyze the estrus behavior of cattle. Although the IoT (Internet of Things)-based wearable wireless sensors such as Lora and BLE are available, it is necessary to install the device on the body of the animal. If one uses these wearable devices, installing them on the tailhead of each cow is necessary. Thus, the cattle are stressed when wearing those devices for twenty-four hours, seven days a week. In addition, the accuracy of the sensor results cannot be checked. Low-cost trackers based on the IoT devices have many adverse impacts on the livestock and farms, including network connectivity, power dependence, higher costs, and adverse health issues.

Computer vision is a more practical approach that does not require attaching any material or a sensor to the cattle being monitored [1]. Vision technology is an integral part of the Fourth Industrial Revolution. With the rapid growth of Industry 4.0, manual monitoring of individual animal behavior on commercial livestock farms is no longer sustainable for the workflow [2]. Currently, there is significant research applying deep learning and computer vision techniques to precision livestock farming and promoting the development of intelligent systems for use on livestock. An essential focus of this research is multiple-object detection and tracking for remotely monitoring numerous animals and capturing their activities.

In a computer vision-based system, the applications of deep learning and neural networks are becoming sophisticated, and their use in models for object detection has increased substantially. Such object detection systems are finding many uses in real-world applications, such as autonomous driving, robot vision, and video surveillance. Popular one-stage object-detection systems include the Regions with Convolutional Neural Network features (R-CNN) [3], You Only Look Once (YOLO) [4,5,6,7,8,9,10,11] and its variants, as well as the Single Shot Multi-Box Detector (SSD) [12]. Rather than a traditional, selective search, the Faster R-CNN is a feature extractor that uses a region proposal network (RPN) [13] to check for the occurrence of an object. As an extension of the Faster R-CNN, the Mask R-CNN [14] is an example of a two-stage deep learning network that performs region segmentation at the pixel level.

From these related studies, we can see that these non-contact monitoring methods have facilitated automation and improved the accuracy of precision dairy farming. The above mentioned deep-learning algorithms work well in detecting objects in static images. While multi-object detection refers to locating several objects belonging to a category of interest within the image, multi-object tracking can be described as tracing the movement of objects throughout a consecutive number of video frames, and consistently assigning individual object IDs.

In recent years, many research studies have sought to adapt deep learning and computer vision techniques to promote the development of smart livestock farming. Farm animals besides cattle have also featured in deep learning applications as part of the Industry 4.0 smart farming. In 2020, a study developed an automatic sheep counting system based on multi-object detection, tracking, and extrapolation techniques [15]. In 2022, a semantic segmentation and counting network was proposed to improve the segmentation accuracy and efficiency of counting pigs, an application requiring sophisticated image segmentation [16]. Another paper proposed a new solution to automatic cow detection and tracking, which optimized cow tail detection and tracking with an improved single-shot multi-box detector (SSD) and Kalman filter [17]. In 2021, a Holstein cattle open dataset was developed [18], but no dataset then existed for black cattle re-identification. This gave us an idea for improving the tracking phase in a farm management system.

We have previously developed a method of detecting and analyzing cattle mounting behavior [19]. We presented an approach to detect and track individual cattle outdoors, using inexpensive color (RGB) surveillance cameras under typical commercial farm conditions. This approach does not use individual cattle identifiers such as RFID due to their impracticality and other concerns. The multi-object tracking algorithm assigns a unique ID to each target, and this ID remains constant throughout the sequence. Before tracking, the detection algorithm obtains the object information in each frame. Therefore, tracking performance mainly relies on the detection results [20,21].

This paper proposes two contributions that are unique to our research. The first is an improved Strong-SORT [22] algorithm featuring a re-identification method for black cattle tracking, and the second is the comparison of the use of an extended Detectron-2 [23] with a customized light-weight tracking algorithm. According to the literature review, multi-object-tracking has never been performed on a black cattle dataset. We implemented a way of tracking cattle through deep learning-based multi-object tracking algorithms. Since the cattle are black and constantly moving, it is hard to rely solely on tracking them by their appearance. This too often results in an ID switch or counting mistake due to occlusion. Since YOLO v5 is a one-stage detector, we used it as our baseline detector. The main contributions of the study are summarized as follows:We compared the use of the cutting-edge detector YOLO v7 with the two-stage detection and instance segmentation of Detectron-2;We developed a customized tracking algorithm after comparing two state-of-the-art MOT algorithms, Deep-SORT and Strong-SORT;We used ResNet 50 as the backbone and modified the Strong-SORT algorithm by building a re-identification network structure for feature extraction in the appearance feature stage to make it more suitable for the black cattle dataset;We built the Re-ID dataset and presented a process for producing a semi-automatic nested dataset;We conducted extensive experiments on a large-scale dataset to validate the proposed approach. The results have indicated that our customized tracking algorithm enables high accuracy in tracking livestock.

The rest of the paper is composed of five sections. Section 2 describes the data used for this analysis and explains the methods applied in this article. Section 3 is the experimental implementation design, and Section 4 describes the experimental results and discussion. Finally, Section 5 concludes this article.

## 2. Materials and Methods

### 2.1. Self-Built Dataset Acquisition

The data used in this research were from the customized dataset supported by the Sumiyoshi Ranch, University of Miyazaki, Japan. These self-collected images are more complex and closer to real-life conditions; therefore, they are ideal for testing the merits of the proposed method. The surveillance camera setup monitored 10 to 20 cattle, as illustrated in Figure 1. Cattle were free to move about in this outdoor environment. The images were continuously captured by a GV-FER5700 fish-eye camera located on top of a barn, providing the best possible view. Data for Japanese black cattle were collected for 24 h a day at 25 frames per second from 25 May to 6 June 2022, for a total of 11 days of raw video. The image sequences also included the walking and lying positions of the cattle. The processing of the dataset was challenging for the following reasons: (1) the cattle changed posture frequently; (2) they were the same variety and colored similarly; (3) the lighting changed continually; (4) a complex background (including trees and ground) was difficult to distinguish from individual cattle. Images acquired in the morning were in a low light, while images at noon were over-illuminated with strong shadows. These lighting problems can cause the deep learning algorithms to inaccurately learn patches or shadows as cattle features. In some rare cases, wet soil cannot be differentiated from the cattle. Our researchers manually annotated 3857 images in the dataset (80% and 20% for training and validation, respectively). We have used an unknown dataset for the testing dataset.

The challenges have largely resulted from the fact that the data were collected in real-world conditions within a commercial farm environment. For example, there were many annotations per image in the dataset because the cattle were often densely packed into a small space. This made separating the cattle very difficult, even using manual annotations. The larger the scale and higher the quality of the data, the more successfully the model can generalize. To address this problem, data augmentation methods using geometric changes, flipping, rotation, clipping, scaling deformation, and affine operations were applied to the dataset. Also, mosaic augmentation was utilized to enhance images. These methods can increase diversity and variation in data samples to a certain extent, as explained in the following Table 1.

When labeling images in the cattle dataset for training, the nested dataset production process featuring the Detectron-2 framework for automatic object detection and segmentation was used. When some objects are difficult to detect, manual annotation is inevitable. Blurred images of small cattle must be annotated manually as they are especially difficult to detect automatically. A combination of manual and automatic annotation will help reduce the human workload. The present study proposes an efficient semi-supervised, nested dataset object detection method, which aims to lower labor costs by reducing the need for human intervention.

In the experiments, over-detection is utilized to prevent gaps in tracking cows, and the over-detected regions are removed manually. Moreover, in creating a dataset in a complex environment, accurate detection of occluded or small cattle is the key to preventing the identification of cattle. For this reason, in this proposed annotation framework, after the automatic detection result from the detector was obtained, detection of the missing cattle began because the small cattle can be occluded or partly obscured behind the bigger cattle, which is the nature of the dataset.

### 2.2. The Proposed System

The proposed system aims at comparing the state-of-the-art deep learning tracking algorithm Strong-SORT with the customized light-weight tracking system.

The proposed cattle detection and tracking system is composed of the following five processing subsystems: (i) video data processing, (ii) cattle detection, (iii) feature extraction, (iv) re-identification (re-ID), and (v) tracking. The re-identification part is only included in the Strong-SORT tracking algorithm. In this part, we proposed the black cattle re-identification dataset as a modification of the Strong-SORT algorithm. An overview of the process components and methods are shown in Figure 2.

### 2.3. The Object Detection

The object detection combines the recognition and localization of the object. It is widely used in various applications, such as intelligent traffic, pedestrian counting, animal recognition, agricultural product pest identification, and defect detection. The Convolutional Neural Network (CNN) is widely used as an important part of computer vision, and object detection technology based on deep learning has also attracted much attention. YOLO (You Only Look Once) is a detection algorithm that pursues both accuracy and speed. It treats detection as a regression problem and can detect objects at fast speeds.

YOLO algorithms have been improving in both speed and accuracy. The YOLO v1 used pre-defined candidate areas to divide the input image into a grid, and each grid can predict multiple bounding boxes. Subsequently, the YOLO v2 adopted many improvements over the YOLO v1, such as normalizing each layer of the network and using a high-resolution classifier. This classifier improved the mAP of the model, but failed to solve the problem of small object recognition. The YOLO v3 introduced a residual module to deepen the network structure. Multi-scale feature information can predict different sizes of objects and improve the accuracy of small object detection. The YOLO v4 and YOLO v5 have further improved in accuracy and speed. Compared with other versions of YOLO, the YOLO v5 is light-weight, and can quickly and accurately detect small objects. The YOLO v7 was released this year, and it includes an instance segmentation module as a branch.

Successful detection is an important precursor to tracking individual cattle. We used two detectors for this purpose: a one-stage and a two-stage detector. Firstly, we used Detectron-2 instance segmentation for subtracting the background and proceeding to the tracking stage. Considering that Detectron-2 outperforms many other detection approaches, we used its output bounding box to train YOLO.

### 2.4. Object Tracking

Object tracking techniques have become a fundamental part of real-time video-based applications that require object correspondence between the frames. According to the literature, recent advances in multi-object tracking (MOT) have been focusing on two different approaches: (1) a tracking by detection approach [24] and (2) a joint tracking and detection approach [25]. Tracking by detection algorithms are used to detect and classify objects before performing the object association, simplifying the process as tracking becomes a matter of associating objects over consecutive frames. Recent research has also employed the Kalman Filter (KF) algorithm [26] as a motion model, which improves the association of objects over time. SORT [27] is a technique consisting of a KF estimate of object states. Subsequently, a year later, other authors have proposed Deep-SORT [28] as an improvement that includes a novel cascading-association step using CNN-based object appearance features.

The data association algorithm combines similarities in the object appearance features with the Mahalanobis distance between object states. The Deep-SORT method achieved a promising frame rate on object tracking benchmarks [29]. Euclidean distances between extracted object appearance features are used to improve the association step. In 2022, Strong-SORT achieved a new state-of-the-art performance on the popular benchmarks like MOT 17 and MOT 20 [30].

Cattle seek shelter according to the weather, which complicates the detection process. Under these conditions, developing techniques for automatically tracking cattle could significantly lower labor costs and improve statistical performance. In this paper, we propose a new customized light-weight cattle tracking approach that can perform under actual conditions on a cattle ranch without affecting the appearance of the black cattle.

### 2.5. Evaluation Methods

To assess the overall performance of our framework, we separately evaluated instance segmentation and multi-object tracking. For Detectron-2 instance segmentation, the mean average precision (*mAP*) is defined as the mean of *AP* across all the categories (*M*), which is shown in Equation (1):(1)$$mAP=\sum_{i=1}^{M}\frac{AP_{i}}{M},$$

For the detection stage, we computed the number of True Positives (*TP*), False Positives (*FP*), and False Negatives (*FN*) over all the test images, and then calculated Recall, Precision rate, and *F*1-score, defined as in the following Equations (2)–(4):(2)Recall=TPTP+FN
(3)$$\mathrm{Precision}=\frac{TP}{TP+FP}$$
(4)$$F1=\frac{2TP}{2TP+FP+FN}$$

If the intersection over union (IoU) threshold was set to 0.5 or 50%, the *mAP* is called maP_0.5 or *mAP*@50.*mAP*_0.5:0.95, meaning that the *mAP* is with the 0.5 < IoU < 0.95 s. The tracking performance measure we used was the multi-object tracker accuracy (*MOTA*). This is the most common metric for benchmarking MOT solutions [31], as it accounts for the three types of error that occur: the false negative (*FN*), false positive (*FP*), and identity switch (IDs), as described in Equation (5). False negatives are defined as objects that are not tracked, false positives are defined as tracked objects which should not be tracked, and identity switches are when two objects that should be tracked swap identities. Fragmentations are defined as the number of times an identity switches from “tracked” to “not tracked”:(5)$$MOTA=\frac{\sum_{t}FN_{t}+FP_{t}+IDSW_{t}}{\sum_{t}GT_{t}}$$
where *t* is frame index, *FN* is false negative, *FP* is false positive, *IDSW* is ID switch, and *GT* is ground truth objects.

## 3. Experimental Design

### 3.1. Experimental Setup

All experiments were conducted on a Windows 10 system with an Intel^®^ Xeon^®^ Gold 6326 CPU @2.90 GHz 16.04, and an NVIDIA RTX a6000 with 32 GB of memory. The proposed cattle instance segmentation model was based on the YOLO version and Detectron-2 implementation used by Pytorch. A Python 3.8.11, TensorFlow 2.5.0, and Keras 2.0.8 with GPU are the requirements. The base model was pre-trained on the Microsoft Common Objects in Context dataset (MS COCO).

### 3.2. Choice of Detection Model

In the choice of the detection model, four models of YOLO v5, three models of YOLO v7, and Detectron-2 were used. Each one was constructed using basic backbone networks with different widths and depths. To select the base model suitable for detecting cows, the transfer learning method was employed, which uses the COCO dataset with pre-trained weights. As the balanced dataset was used, no over-fitting occurred during the training process. Images with different resolutions were introduced into models of different batch sizes for training. Figure 3 describes the training and validation loss of YOLO v5 on the cattle dataset. Figure 4 describes the results from the Detectron-2 training.

### 3.3. Fitting the Detection Model in the Tracking Process

In this tracking process, the systems describe two tracking algorithms: (1) Modification of the Strong-SORT-based re-identification dataset, and (2) the proposed customized light-weight tracking algorithm.

#### 3.3.1. The MOT Network for Automatic Tracking

In this section, the modified Strong-SORT-based re-identification approach [32] is described. The cattle tracking algorithm presented in this paper has a higher tracking success rate than the original YOLO v5 with the original Strong-SORT model, and is more effective and stable in tracking cows. The cattle are detected using the YOLO v7 network from the detection model, and sent to the Strong-SORT. After that, the algorithm updates the positions of the tracked targets.

Strong-SORT is an improved version of the Deep-SORT multi-object tracking algorithms, and it includes an association method, which improves the accuracy of tracking objects for long periods of time and reduces the rate of ID switching. Strong-SORT uses an appearance branch, which is a stronger appearance feature extractor than Deep-SORT. By taking ResNet 50 as the backbone and pre-training on the DukeMTMCreID [33] dataset, Strong-SORT can extract more discriminative features than Deep-SORT.

A Strong-SORT tracking algorithm contains six main steps. The first step is object detection. The second is pre-processing and threshold selection. The third is feature extraction, in which the Re-ID model extracts appearance features. The fourth step involves associating the data with that in the previously detected frame. The fifth is track management, which includes updating the Kalman filter, as well as initializing and deleting tracks. After the fifth step of post-processing, the tracking results are obtained. Figure 5 shows the process for cattle tracking with Strong-SORT.

In the proposed modified Strong-SORT algorithm, only the black cattle re-identification part is improved. After cattle are detected by YOLO v5 and YOLO v7, the cattle regions are forwarded to the Strong-SORT algorithm. Following this, the cattle re-identification process is initialized, and cattle are identified. The re-identification step is explained in the next section. Finally, the bounding boxes with IDs are obtained.

#### 3.3.2. The Cattle Re-Identification Dataset

The original Strong-SORT was trained on the Person Re-ID Dataset to extract appearance features. People are distinguished by clothes of various colors and patterns. The cattle in our dataset are uniformly black, which is the main challenge in re-training the Strong-SORT feature extractor. Therefore, for the extraction of appearance features of the cattle, the re-identification is performed. A sample of the cattle Re-ID dataset is shown in Figure 6.

Building the cattle re-identification dataset involves other challenges as well, including intra-class variations, illumination changes, low resolution, and occlusion. The person dataset contains 751 different classes. The cattle Re-ID dataset consisted of 70 cattle identities, each with an average of 240 images, totaling 16,800 images (80% for training and 20% for testing). All annotations were resized to 128 × 128 for processing by a Convolutional Neural Network (CNN), describe in Table 2. The proposed unsupervised re-identification network removes the need for annotated datasets. To build the cattle Re-ID dataset, separating each cow into each group is a necessary step. After the detection stage, cattle images are saved from the YOLO detector. These are automatically split using a traditional image processing-thresholding method. Based on the input images, we defined the optimal threshold parameters to split the group automatically. The best threshold values for each image are selected by analyzing all of the images. After that, the automatic splitting process is performed. Subsequently, each same image is kept in the same group as described in Figure 7. To the best of our knowledge, it is the first work to propose unsupervised Re-ID models for multi-object tracking.

In the experiments, the Adam optimizer (Adaptive Moment Estimation) [34], which is a computationally efficient optimizer, selected a learning rate of 1 × 10^−5^ and a batch size of 128. While training with the CNN, the epochs stopped at 100, when the training accuracy reached 95%. When the training reached 200 epochs, the system overfitted the data. In order to facilitate training the Re-ID model and avoid overfitting the model, the best validated model in the first 100 epochs was used for testing. We used transfer learning, which can improve the generalization ability of the model [35]. The original weight was applied and it was used to pre-train the improved re-identification network.

After completing the process of re-identification, the tracking stage of Strong-SORT is processed.

The proposed modified Strong-SORT tracking stage has the following states with the parameters shown in Table 3: (1) Tentative State: new tracks that have not been confirmed; (2) Confirmed State: tracks that have been validated as confirmed; (3) Deleted State: tracks that will be removed at the end of the iteration. This tracking stage is from the fourth and fifth steps of the Strong-SORT as described in the previous Section 3.3.1. In Table 3, *N_INIT* is the number of frames that a track remains in the initialization phase which is the same stage as the Tentative State; *NN_BUDGET* is the maximum size of the cattle store in the database. The *MAX_AGE* determines the maximum number of frames for which an ID will be kept alive (or active) without any valid associations, which is the same state as the Confirmed State. If an ID is not associated with any detections after the *MAX_AGE* number of frames, then the ID is deleted which is the same stage as the Deleted State. This helps to account for some missed detections (false negatives) by the detection model. The Strong-SORT tracking results are described in Section 4.

#### 3.3.3. The Proposed Customized Light-Weight Tracking Mechanism

In this section, to compare with the modified Strong-SORT tracking algorithm, we proposed a customized light-weight tracking algorithm which is composed of three primary stages, as shown in Figure 8. Detection is the first stage, adopted from Detectron-2. The second stage is matching, in which the detections in the current frame with the locations of detections in the previous frames are associated (Figure 9). The third stage is re-matching, in which new detections are re-assigned. Section 3.3.4 and Section 3.3.5 describe these stages in more detail.

#### 3.3.4. Matching

After completing the detection stage, we associate the detections in the current frame through a matching method that is based on the target’s previous locations in previous frames. Once a cow is detected, the position in the next frame can be calculated according to Algorithm 1. When some cattle move quickly, the IoU may be too low between frames, therefore, affecting the matching stage. Similarly, if parts of the cattle are occluded, it affects the IoU and other components of the score.

#### 3.3.5. Re-Matching

The proposed system matched the detection pairs according to the matching method. The system defined detections in previous frames, which have *N* paired detections, and the current new detections without pairs to the previous detections are called new detections. Every 30 frames, the system recalls the matching method, which is a re-matching with the previous detection and the new detection. At frame *t* in the re-matching stage, the new detections will re-match with all the previous detections. Once one of the new detections are successfully re-matched with a previous detection, the new detections will be assigned the previous detection IDs, which means the correct IDs are assigned. If those new detections are not re-matched with previous detection IDs, the system assigns them with new tracking IDs. The system sets a maximum number of frames after which the IDs that are no longer in the detection process are deleted. Algorithm 1 describes the customized light-weight tracking algorithm.
**Algorithm 1.** Cattle Tracking**Input:** Bounding Boxes and mask-regions for each cow from Detectron-2 detection
**Output:** IDs, the list of cow IDs1: **Function:** Cattle_Tracking (Bounding Boxes, Mask Region):2: Calculate Euclidean Distance for Aspect Ratio, Centroid values, and differences in Mask Area for each cow from frame to frame.3: Perform the following steps for each cow4. Assign each cow a tracking ID, calculating the distance between the previous and current frames using the cost function (8):
(6)$$L_{dist}=\sum_{i=t}^{N}\frac{\sqrt{{\left(x_{i}-x_{i-1}\right)}^{2}+{\left(y_{i}-y_{i-1}\right)}^{2}}}{N}$$
(7)$${Ar}_{dist}=\sum_{i=t}^{N}\frac{\sqrt{{\left(Ar_{i}-Ar_{i-1}\right)}^{2}}}{N}$$
(8)$$COST=L_{dist}+Ar_{dist}$$

where Ldist is the Location Distance; *t* is the current frame; *x*, *y* is the bounding box location; Ldist is the Average Euclidean Distance from current to *N* frames; *Ar* is the Aspect Ratio (Ar = w/ℎ); and COST is the combination of two distances (Location Distance and Aspect Ratio Distance).5: Calculate to check the tracking database for the same detected object. 6: Compute the same object-to-tracking ID values from the tracking database. 7: Assign new tracking IDs for new cattle in the tracking database.8: **Return IDs**9: **End Function**

The system utilized two costs for matching the ID, the location and the aspect-ratio distances; both are Euclidean distances. In this stage, the system associates detections in the current frame with the object’s previously detected locations. The location distance is defined in Equation (6). When matching two detections, a match is more likely when their union is smaller, and has a larger overlapping area between their corresponding bounding boxes. A match is also more likely with a higher IoU score, as described in Equation (9):(9)$$\mathrm{IoU}=\frac{Overlapping\;Area}{Union\;of\;Area}\;<=1.0$$

To test the performance of the algorithm, we first experimented with 15-min video clips and then integrated them into a one-hour video to see the results described in Section 4.

## 4. Experimental Results and Discussion

### 4.1. Detection Results

In this section, the detection and tracking results of the experimental results are described. To evaluate the detection performance [36], the precision, recall, mAP of IOU = 0.5:0.95 and 0.5, and inference time are compared in each of Table 4, Table 5, Table 6, Table 7 and Table 8. Some of them require high computation power, while others are more efficient in the processing time. Among these models, the YOLO v5 is the best in both precision, accuracy, and efficiency in training time. In addition, various CNN baselines are tested for the Detectron-2 feature extraction process. Various ResNet [37] model sizes of the tested baselines are compared in Table 7.

### 4.2. Tracking Results

To evaluate the performance of long-term tracking, a long video of 60 min is used to test the model’s performance. For a comprehensive analysis of the proposed tracking method, Table 8 shows the experimental setup and results. Table 9 provides an analysis of tracking 20 cattle using 15-min video segments. Sometimes, the ID switching issue occurred because the occlusion between cattle often resulted in false negative detections. If a lost track reappeared, it was not associated with the correct ID. This was due to the detection limitations and challenges. Table 9, Table 10 and Table 11 show the execution time (inference time), which is the overall processing time for each one-hour video for retrieving the results.

Figure 10 shows the results of using one-stage and two-stage detectors. In the result, the YOLO v7 outperforms the other one-stage detector, which was tested on the same video clip. The Faster R-CNN object detector achieved especially poor results in detecting small cattle which were further away from the camera.

Figure 11 and Figure 12 show the qualitative results of using the YOLO v7 with the modified Strong-SORT, compared with using the YOLO v5 with the Deep-SORT, which contains the largest number of cattle that were recorded in extreme lighting conditions. Each video shows the typical cattle movement in the ranch where the cattle were detected and tracked properly. Despite the inclusion of many cattle, the system tracked accurately. This shows the proposed system is robust, even with a high density of cattle moving frequently and with the overlapping images.

The knowledge of social interactions between cattle is fundamental to enhancing farming conditions and promoting animal welfare. The main goal of the proposed system was to observe interactions between cattle and identify behavior patterns associated with estrus. The system first focused on proximity contacts between at least two cattle and then constructed a trajectory map using this information to highlight the individual patterns of movement. Figure 13 provides an example of identifying behavior in a test video sequence with cattle, which can also be used for extracting a distinct contact period by restricting the frame and ID information.

## 5. Conclusions

This study compares various combinations of detectors and tracking methods in the effort to improve both functions. For tracking cattle, the system also compared the use of a modified Strong-SORT with the proposed customized light-weight tracking system. As the main contribution of this study is to develop a complete end-to-end system that can process raw RGB images in the accurate detection and tracking of black cattle. This system has overcome the challenges of occlusion and cattle density in real-world conditions on a farm, with a single RGB camera in a fixed location.

In interpreting the findings of this study, we confirmed that the system achieved the tracking of the cattle even when tracking is interrupted by occlusion or changes in illumination. Long-term tracking can provide continuous and automatic monitoring of the health and welfare status of the animals. Moreover, this tracking methodology can be used for other species besides cattle, and for detecting behavior such as lameness and mounting. With improved detection of behavior in estrus, the system can pinpoint the optimal time for artificial insemination. One challenge of this system is the computing time required for the real-time processing of the video data. The comparison between the one and two-stage detection algorithms showed that the one-stage model provided a higher inference speed.

The proposed system compares favorably with other cattle detection and identification methods. The accuracy achieved in [38] was only 85.4%. This study used a dataset with 5042 images of full cattle bodies and featured the Mask R-CNN detection method using VGG 16 as the baseline backbone. In contrast [39], achieved 73% accuracy using the YOLO v3 on 11,754 frames and Dark Net as the backbone. In [40], an accuracy of 94% was achieved using the deep learning Mask R-CNN on 750 images. In [41], a Faster R-CNN, this one-stage detection network achieved an accuracy of 89.1%; however, only 43 images were used. An accuracy of 89.95% was achieved in [42], which used an SDAE detection method on 1000 images with CNN as the baseline network. In paper [43], the authors used CNN and LSTM-based identification methods, achieving an accuracy of 88%. In paper [44], the authors used Temporal Segment Networks (TSN) identification methods and achieved an accuracy of 84.4%. It is evident that the proposed method is a significant improvement over other dairy cattle detection algorithms, and the feasibility of detecting and tracking black cattle is validated at a significant level.

Compared to the aforementioned studies, the proposed system in the present study used a black cattle dataset. The aim of constructing this dataset is to meet the formidable challenge of detecting and tracking black cattle without the additional infrastructure. However, despite the improved performance of the method, the system could still not fully overcome the ID switching issue, which is one of the most frequently stated problems in MOT. In this study we found that the ID switching issue mainly resulted from occlusion, and from when the cattle left the camera’s field of view. Through the combination with other systems, we are confident that our work will soon enable the more intelligent monitoring of livestock.

We developed a system based on a comparison of various combinations of deep learning detection and tracking technologies applied to a black cattle dataset. The ability to track individual cattle with deep learning is the key to creating a full system that can provide all the information required for farm management, including the early detection of the optimal time for artificial insemination. The experiment results show that the proposed method allows the tracking of cattle for up to 1 h, with a MOTA of 96.88% without the use of additional hardware and using only a standard RGB camera. The tracks derived from this system can be used to calculate the behavioral metrics for future work. The proposed system also introduced different deep learning-based trackers. In summary, we have proven that our proposed customized light-weight tracking algorithm performs the best tracking accuracy and is better than the modification of Strong-SORT in our black cattle dataset.

In future research we plan to make our tracker more stable by improving the optical flow and addressing the variability in light intensity. A more significant challenge lies in effectively implementing deep learning methods. Future work will involve testing the strategy on more complex trackers and extending our work to apply in real-time cattle monitoring.

## Figures and Tables

**Figure 1 sensors-23-00532-f001:**
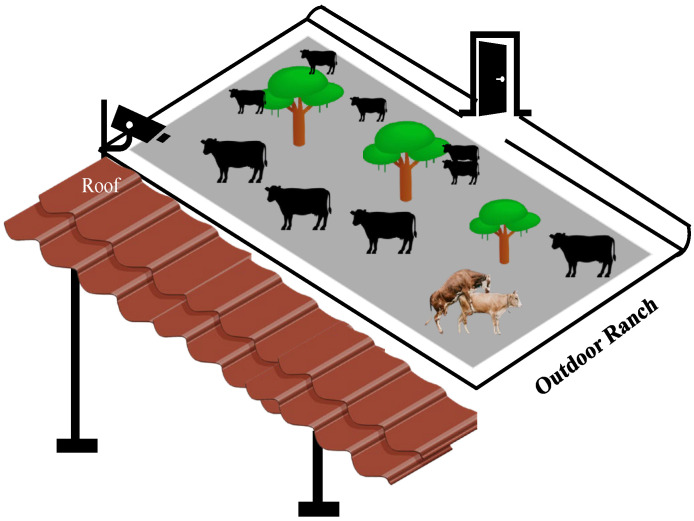
An illustration of the ranch layout.

**Figure 2 sensors-23-00532-f002:**
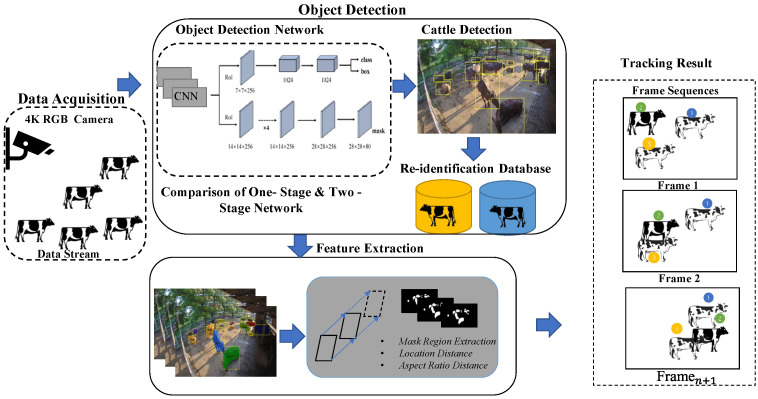
The complete steps of the proposed system.

**Figure 3 sensors-23-00532-f003:**
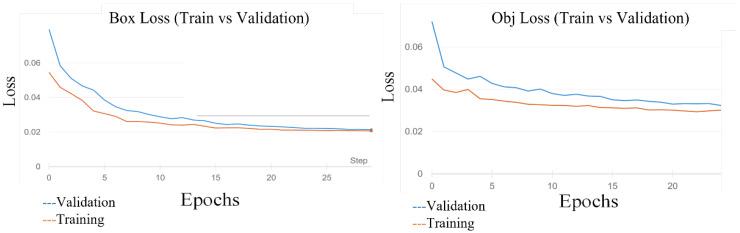
The training and validation loss of YOLO v5 on the cattle dataset.

**Figure 4 sensors-23-00532-f004:**
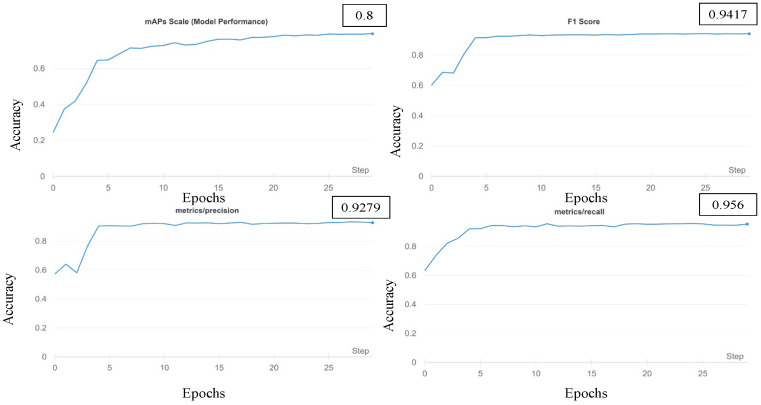
The results from the Detectron-2 training.

**Figure 5 sensors-23-00532-f005:**
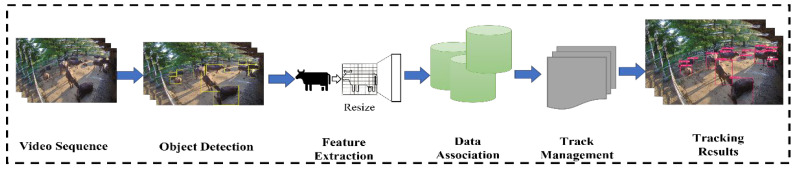
Cattle Tracking with Strong-SORT.

**Figure 6 sensors-23-00532-f006:**
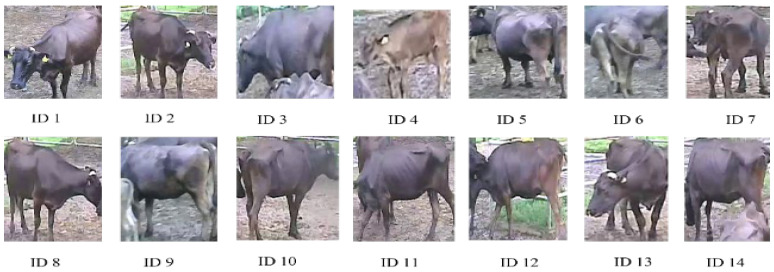
A sample of the cattle Re-ID dataset.

**Figure 7 sensors-23-00532-f007:**
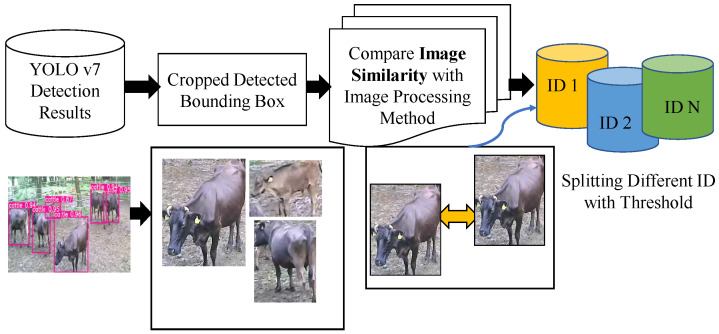
Building the cattle re-identification dataset with the thresholding method.

**Figure 8 sensors-23-00532-f008:**
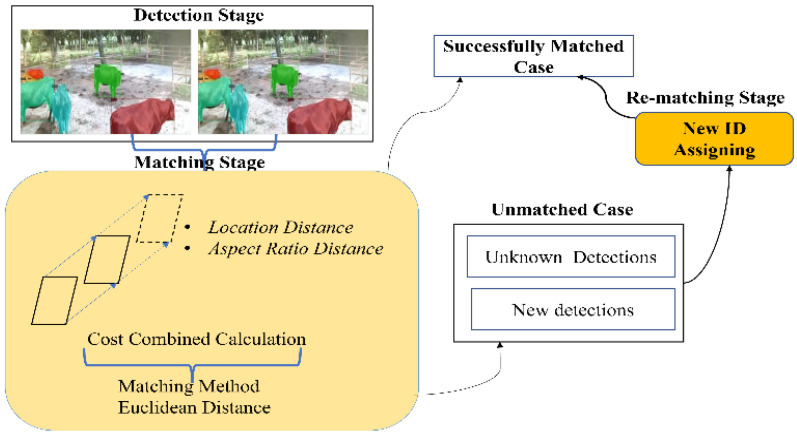
The proposed customized tracking algorithm.

**Figure 9 sensors-23-00532-f009:**
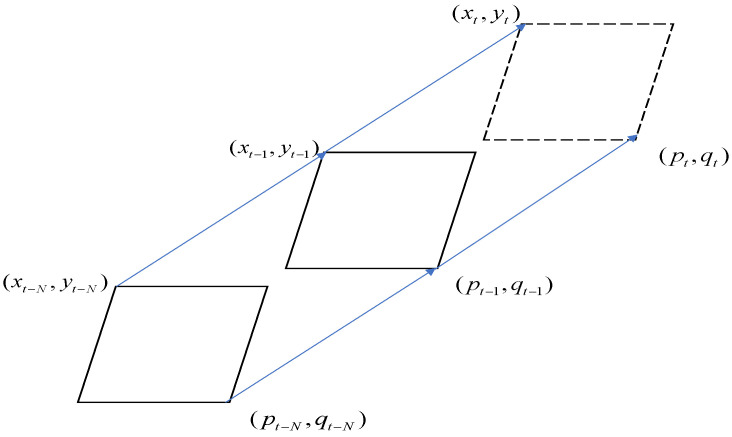
The explanation of the customized tracking algorithm, where (x, p) and (y, q) are the left upper point and the right upper point, respectively. The index t-N represents the N frame previous detection; t-1 represents the previous detection; and t means the current frame location.

**Figure 10 sensors-23-00532-f010:**
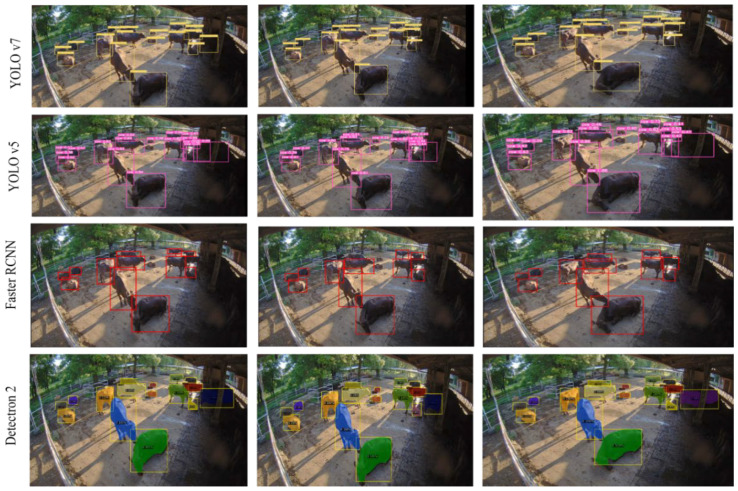
A comparison of detection results between the one- and two- stage detectors.

**Figure 11 sensors-23-00532-f011:**
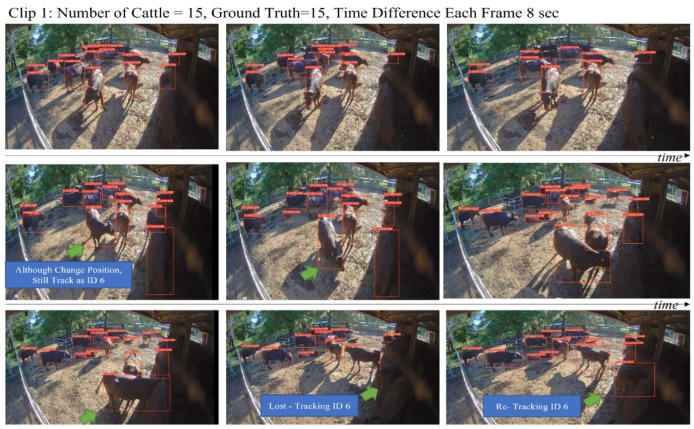
An analysis of the YOLO v7 with the modified Strong-SORT results.

**Figure 12 sensors-23-00532-f012:**
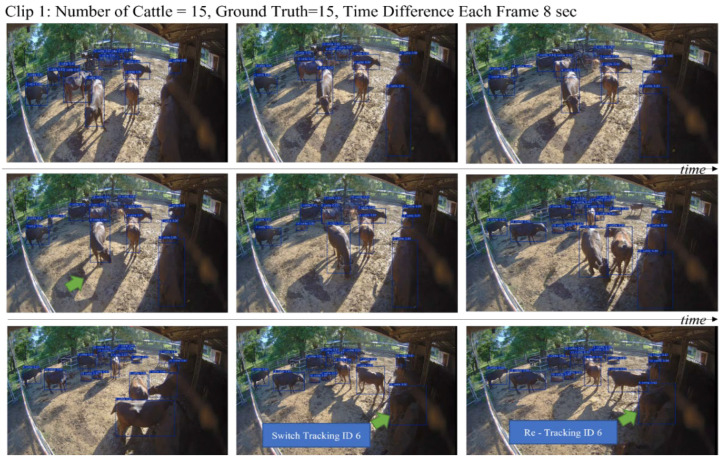
An analysis of the YOLO v5 with the Deep-SORT results.

**Figure 13 sensors-23-00532-f013:**
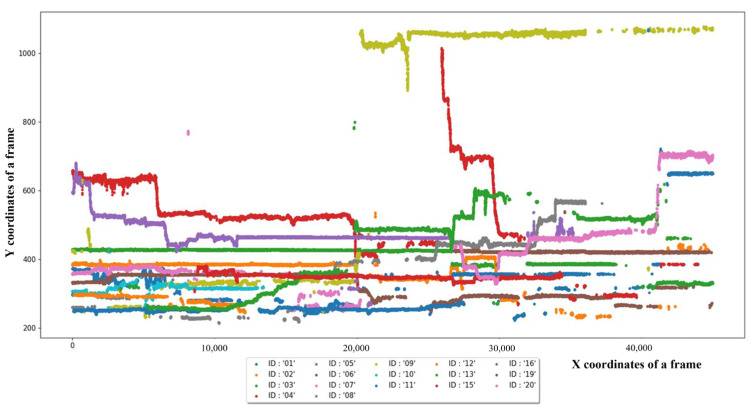
An analysis of tracking 20 head of cattle with the Detectron-2 with the light-weight tracking mechanism.

**Table 1 sensors-23-00532-t001:** The explanation of the dataset.

Dataset	Number of Videos	Number of Images	Image Enhancement Method	Image Resolution
Training set	30	3086	Mosaic augmentation	2560 × 2048
Validation set	5	771	-	2560 × 2048
Total	35	3857	-	2560 × 2048

**Table 2 sensors-23-00532-t002:** An overview of the CNN architecture used to learn the cattle re-identification.

Layer No.	Name	Patch Size/Stride	Output Size
1	Conv 1	3 × 3/1	32 × 128 × 128
2	Conv 2	3 × 3/1	32 × 128 × 128
3	Max Pool	3 × 3/2	32 × 64 × 64
4	Residual 4	3 × 3/1	32 × 64 × 64
5	Residual 5	3 × 3/1	64 × 32 × 32
6	Residual 6	3 × 3/2	64 × 32 × 32
7	Residual 7	3 × 3/1	128 × 16 × 16
8	Dense 8	-	128
9	Batch Normalization	3 × 3/1	128
10	Relu	3 × 3/1	128

**Table 3 sensors-23-00532-t003:** Strong-SORT parameters and settings.

Parameters	Value
MAX_DIST	0.9
MAX_IOU_DISTANCE	0.8
MAX_AGE	1500
N_INIT	1
NN_BUDGET	100

**Table 4 sensors-23-00532-t004:** The detection results of the different YOLO v5 models.

Model	Precision %	Recall%	mAP% (IOU = 0.5:0.95)	mAP% (IOU = 0.5:)	Inference Time (s)
YOLO v5s	0.999	0.985	0.928	0.995	3500
YOLO v5m	0.999	0.985	0.936	0.994	3880
YOLO v5n	0.998	0.985	0.884	0.993	3500
YOLO v5l	0.998	0.985	0.994	0.947	4548

**Table 5 sensors-23-00532-t005:** The detection results with the different YOLO v7 models.

Model	Precision %	Recall%	mAP% (IOU = 0.5:0.95)	mAP% (IOU = 0.5:)	Inference Time (s)
YOLO v7	0.999	0.985	0.871	0.997	18,292
YOLO v7-X	0.999	0.985	0.891	0.997	7998
YOLO v7-W6	1.000	0.985	0.896	0.997	7815

**Table 6 sensors-23-00532-t006:** A comparison with the state-of-the-art one-stage detector.

Model	Precision %	Recall%	mAP% (IOU = 0.5:0.95)	mAP% (IOU = 0.5:)	Epoch	Inference Time (s)
Faster RCNN	0.835	0.907	0.953	0.987	1500	2724
YOLO v5s	0.999	0.985	0.928	0.995	100	3500
YOLO v7s	0.999	0.985	0.871	0.997	55	18,292

**Table 7 sensors-23-00532-t007:** A comparison of the different ResNet backbones for Detectron-2.

Methods	Depth	Iteration	Inference Time (s)	Validation Accuracy
ResNet 18	18	1000	3500	0.894
ResNet 50	50 + FPN	1000	3800	0.916
ResNet 101	Original	800	2500	0.914
		900	3200	0.921
		1000	3340	0.948
		1100	3500	0.948
	C4 3x	1000	4200	0.929
	32 × 8d FPN 3 ×	1000	4300	0.948

**Table 8 sensors-23-00532-t008:** The performance of the two-stage detector for comparison.

Methods	mAP @ 0.5 (%)	Inference Time (s)
Detectron-2	0.927	6300

**Table 9 sensors-23-00532-t009:** A comparison of the Multi-Object tracking methods.

Methods	MOTA	ID-Switch	Inference Time (s)
YOLO v5 + Deep-SORT	92.52%	10	1260
YOLO v7 + Strong-SORT	95.32%	7	1800
Ours (YOLO v7 + Modified Strong-SORT)	96.88%	4	3600
Ours (Detectron-2 + Tracking)	96.88%	4	3900

**Table 10 sensors-23-00532-t010:** Details of the counting results for each video clip.

Videos	#Cattle Ground Truth	#Tracked Cattle	MOTA (%)
Clip 1	20	19	98
Clip 2	21	20	98
Clip 3	22	21	98
Clip 4	21	21	100
Clip 5	25	23	97

**Table 11 sensors-23-00532-t011:** The execution time of each module in the proposed tracking system.

Modules	Inference Time (s)
Detection module	1700
Tracking module	7200
Total	8400

## Data Availability

The data presented in this study are available on request from the corresponding author.

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
