# Peer review of "Comparing State-of-the-Art Deep Learning Algorithms for the Automated Detection and Tracking of Black Cattle"

_sensors, 2023, doi:10.3390/s23010532_

Round 1
Reviewer 1 Report
The authors propose application of deep learning technologies to a black cattle tracking dataset, to provide all the information required for farm management, including the early detection of the optimal time for artificial insemination.
The authors provide a overview of evolving trends in the field. However, there is a lack of a certain number of works that deal with the identification of individual cattle using machine/deep learning methods, whether it is about recognizing the head, tail, other characteristic parts or the whole body.
On the other hand, the authors made a considerable effort to mention their works (self-citation), which even led to references 17 and 18 being exactly the same:
17. Noe, S.M., Zin, T.T., Tin, P. and Kobayashi, I., 2022, March. A Deep Learning-based solution to Cattle Region Extraction for 658 Lameness Detection. In 2022 IEEE 4th Global Conference on Life Sciences and Technologies (LifeTech) (pp. 572-573). IEEE. 659
18. Noe, S.M., Zin, T.T., Tin, P. and Kobayashi, I., 2022, March. A Deep Learning-based solution to Cattle Region Extraction for 660 Lameness Detection. In 2022 IEEE 4th Global Conference on Life Sciences and Technologies (LifeTech) (pp. 572-573). IEEE.
Related to the experiments themselves, all the steps of the experiments are recounted in detail. However, in some experiments it is not clear whether the data is divided into training-validation-test datasets or perhaps only training-validation datasets? If different datasets were not used for validation and testing, the question can be raised how the hyperparameters were determined. Or maybe the same dataset was used both for determining the hyperparameters and for the final testing?
Reviewer 2 Report
This paper proposes to apply state-of-the-art deep learning-based Multi-Object Tracking (MOT) algorithms to track cattle by using a RGB camera. Results presented by the authors seem to be very accurate when analyzing a sample of video footage.
I suggest a revision based on the following points in order to improve the quality of the paper:
1.- The authors propose a solution for a given problem: tracking and identification of cattle. From my point of view there are simpler, fast and more reliable solutions such as the use of cheap trackers based on Lora, BLE or other protocols, which are already available in the market. The authors are asked to recognize this fact and develop in the manuscript, providing a suitable explanation why they follow this route instead of using trackers.
2.- It is not clear if the results presented can be applied on-line in real-time or if they require some processing before an identification result is obtained.
3.- In the tables of results inference times are presented. Please define the meaning of “Inference time” in the manuscript. Is it the time to process the video sequence? If so, which length of video sequence is taken as the base to determine the inference time?
4.- Why inference times range from some seconds to thousands of seconds? Is this proposal practical in the case of several thousand of seconds? Please develop in the paper.
5.- The authors are suggested to include a paragraph explaining the feasibility to apply this system for an on-line real-time identification and tracking of cattle.
6.- I still think that it is more practical and efficient to use low-cost trackers based on the Internet of Things. You must convince the readers of this solution.
I hope this revision can help the authors to improve the quality and readability of the paper.
Reviewer 3 Report
In the paper Comparing State-of-the-art Deep Learning Algorithms Used for 2 Automated Detection and Tracking of Black Cattle, the authors proposed an approach to detect and identify individual black cattle in the herb. Although object detection is a classical problem in computer vision and is well handled by deep learning, tracking and identifying animals with similar colors and features are still challenging. This manuscript proposes a combination of the object detection and ID tracking approach to meet the challenge. It also compares the performance of different algorithms on the problem. The manuscript attempts to propose the approach but fails present it clearly and makes the readers difficult to understand. It should be improved either in terms of organizing the paper or the English writing. The following issues need to be addressed.
1. There is no clear logic in presenting the proposed approach. The manuscript attempts to put too much information into it but lacks a clear order and strategy. As a reader, I was puzzled after reading the manuscript and did not understand the data flow of the system architecture, etc. I am not saying it is completely messy but not clear enough. To my understanding, as a paper on the application of deep learning, one would organize the structure by five steps: 1. An introduction to the existing technologies. 2. The challenge/limit of the existing approaches on the studied application. 3. The changes one makes to meet the challenges. 4. The overview of the proposed system and the input-output of each section. 5. The experiments and the analysis. The issues in the manuscript are
a). The manuscript mixed the first four steps and states the existing approach and authors’ increments alternatively; it makes the reader difficult to recognize the authors’ contributions.
b). Moreover, the data flow is critical when describing a large system, but the manuscript is not organized in an input-output engineering way. E.g., there are many places to describe the dataset: section 2.1, section 2.3, section 2.4, and section 3.5. These scattered descriptions make the readers’ mind jump back and forth and
c). There should be only one flow chart in a paper and the others can be the detailed extension of the overview one. But the manuscript has flow charts in Fig. 2, Fig. 5, Fig. 7, and Fig. 8. Furthermore, the relation between the charts is not clear, and the interfaces (input-output) of the blocks in the charts are not specified.
2. The manuscript also has many language problems.
a). There are too many redundant descriptions and words. E.g., many statements in section 5 and section 6 overlap. Please see other examples in the strikethrough comments.
b). Sentences have grammatical errors, they either have multiple verbs or no verb, or the usage of the logic conjunctions is wrong, etc. And the statements are confusing. Please see the sections highlighted in green.
In total, the biggest problem of the manuscript is lacking a logical structure. I did not get a clear and organized system by reading through the paper.

Round 2
Reviewer 1 Report
The authors substantially improved their manuscript by adding details.
Author Response
As the reviewer 1 said "the authors substantially improved their manuscript by adding details" in this second round revision, we will not update the attachment file.
Reviewer 2 Report
The authors have replied my questions
Author Response
As the reviewer 2 said "The authors have replied my questions" in this second round revision, we will not update the attachment file.
Reviewer 3 Report
Thanks for the authors’ reply.
This version is improved in terms of language and logic in the first half of the entire paper. I can follow up the authors’ descriptions up to section 2, but after that, it is difficult to understand again. Please pay attention to the following problems.
1. The “results” section is usually put at the end of the main body of a paper. Section 3 mixes the results and the proposed methods and makes it confusing. Please pay attention to the order of the paragraphs; put them in a logical order.
2. The first revision seems messed something up in section 4. Please read it carefully before resubmission.
As a general comment, when I read a paper, I expect it organized as a story; it has a start, development, result, and conclusion. The order of the sections and the paragraphs does matter, otherwise, the readers would be confused. Also, I expect the connection between the sections of the story, e.g., I see a person called “three states”: (1) Tentative State, (2) Confirmed State, (3) Deleted State in section 3. I expect this person promotes the story development in the results section. Did he/she make the story better? Did he/she meet someone else? Unfortunately, I didn’t see this person in the following story anymore. On the other hand, I never see a person called “matching score” in line 414, but he suddenly appears and plays a role in the story. In the paragraph of line 372, the story of table 8 is developing, but in the next paragraph (line 379), a story of "appearance feature" and "distance" gets involved, and the paragraph of line 390 is back to table 8, the "appearance feature" and "distance" are forgotten. These are just three examples; please don’t only rely on and limited in my comments and make the story better.
If the first half and the second half of the paper are written by different authors, please organize them as a whole. Also, please carefully revise and improve the language of the second half (sections 3, 4, and 5).
Please see the detailed comments in the attachment.
Thank you.

Author Response
"Please see the attachment."

Round 3
Reviewer 3 Report
This version is significantly improved. There are still some minor issues in the manuscript; it should be good to go after the correction. Please pay special attention to lines 383-385 and line 523. See the attachment for more details. Thanks.

Author Response
"Please see the attachment."
